## Effectiveness of two doses of Euvichol-plus oral cholera vaccine in response to the 2017/2018 outbreak: a matched case–control study in Lusaka, Zambia

Cephas Sialubanje [1] , Muzala Kapina,[2] Orbrie Chewe,[3,4]
Belem Blamwell Matapo,[5,6] Albertina Moraes Ngomah [7] Brittany Gianetti,[8]
William Ngosa,[7] Mpanga Kasonde,[9] Kunda Musonda,[4,10] Modest Mulenga [11]
C Michelo,[12] Nyambe Sinyange,[13] Patricia Bobo,[4] Khozya Zyambo,[14]
Lucy Mazyanga,[2] Nathan Bakyaita,[5] Victor M Mukonka[2,15]

For numbered affiliations see end of article.

**Correspondence to**
Dr Cephas Sialubanje; csialubanje@yahoo.com

## ABSTRACT

**Introduction** Zambia experienced a major cholera outbreak in 2017–2018, with more than 5905 cases reported countrywide, predominantly from the peri-urban slums of Lusaka city. The WHO recommends the use of oral cholera vaccines (OCVs) together with traditional control measures, including health promotion, provision of safe water and improving sanitation, in cholera endemic areas and during cholera outbreaks. In response to this outbreak, the Zambian government implemented the OVC campaign and administered the Euvichol-plus vaccine in the high-risk subdistricts of Lusaka. Although OCVs have been shown to be effective in preventing cholera infection in cholera endemic and outbreak settings, the effectiveness of the Euvichol-plus vaccine has not yet been evaluated in Zambia. This study aimed to determine the effectiveness of two doses of OCV administered during the 2017/2018 vaccination campaign.

**Methods** We conducted a matched case–control study involving 79 cases and 316 controls following the mass vaccination campaign in the four subdistricts of Lusaka (Chawama, Chipata, Kanyama and Matero). Matching of controls was based on the place of residence, age and sex. Conditional logistic regression was used for analysis. Adjusted OR (AOR), 95% CI and vaccine effectiveness (1-AOR) for two doses of Euvichol-plus vaccine and any dose were estimated (p<0.05).

**Results** The AOR vaccine effectiveness for two doses of Euvichol-plus OCV was 81.0% (95% CI 66.0% to 78.0%; p<0.01). Secondary analysis showed that vaccine effectiveness for any dose was 74.0% (95% CI 50.0% to 86.0%; p<0.01).

**Conclusion** These findings show that two doses of Euvichol-plus OCV are effective in a cholera outbreak setting in Lusaka, Zambia. The findings also indicate that two doses are more effective than a single dose and thus support the use of two doses of the vaccine as part of an integrated intervention to cholera control during outbreaks.

## INTRODUCTION
### Background
Cholera is a major public health problem with an increasing global burden.[1] Current

## STRENGTHS AND LIMITATIONS OF THIS STUDY

⇒ Use of a matched case–control study design and collection of data soon after introduction of the vaccine minimises confounding and increases internal validity of the study.
⇒ Use of a pretested electronic data collection tool (Open Data Kit) and trained data collectors minimised bias due to measurement error.
⇒ Selection of cases from the line list with confirmed diagnosis limits bias due to misclassification.
⇒ Use of the vaccination certificate/card to confirm cases' and controls' vaccination status reduced recall and misclassification bias.
⇒ A case–control study is not a randomised controlled trial, and evidence from such observational studies is considered relatively weak.

estimates indicate that 1.3 billion people are at risk of the disease in endemic countries, resulting in 2.46 million cases and 91 000 deaths worldwide per annum.[1 2] The low-income and middle-income countries are disproportionately affected and majority (60%) of cases occur in sub-Saharan Africa, with some countries reporting a case fatality rate of 7.5%.[1 2] Zambia is one of the sub-Saharan African countries with a high burden of cholera, with over 29 cholera outbreaks recorded since 1977[2] and 34 950 cases reported between 2008 and 2017, the highest number of cases (17 348) having been reported in 2010 from 33 districts around the country.[3]

The most recent outbreak in the country (6 October 2017 to 18 May 2018) was reported from all the ten provinces of the country and recorded 5905 cases.[3] Although all the provinces were affected, past outbreaks have predominantly occurred in the peri-urban

slums of Lusaka city, Copperbelt cities and towns, as well as in fishing camps of the Northern, Luapula, Southern and Central Provinces,[3] attributed to poor access to safe water and sanitation facilities.

The Zambian Ministry of Health (MoH) has historically used a multi-sectorial approach that engages various relevant ministries and cooperating partners[4] to implement various interventions including health promotion, provision of safe water and improving sanitation facilities. Though important, these interventions have not been effective in preventing and controlling outbreaks in the country; they are long-term and extremely expensive, requiring adequate infrastructure and skilled personnel.[5 6]

To prevent and control cholera outbreaks, the WHO recommends use of oral cholera vaccines (OCVs) in conjunction with traditional control measures in cholera endemic areas and during cholera outbreaks.[7] In line with this recommendation, the Zambian government used vaccination programmes in response to the 2016 and 2017–2018 outbreaks in Lusaka district[8 9] and administered Shanchol vaccine and Euvichol-plus vaccine (uBiologics, Seoul, Korea), respectively. Both Dukoral (WC-rBS) & Shanchol and Euvichol-plus & mORCVAX (BivWC) OVCs have been found to be effective in preventing cholera infection in cholera endemic and outbreak settings.[8 9] Further, evidence has shown that two doses of Shanchol provide a cumulative efficacy of 65%–74% over a 5-year period in cholera endemic settings.[10 11] The vaccine has been shown to confer even higher short-term protection during cholera outbreaks.[12–14] For example, during the 2016 outbreak in Lusaka, Zambia, a single dose of Shanchol vaccine provided 88.9% effectiveness in preventing cholera infection.[15]

### Objectives

While Euvichol-plus vaccine has been shown to have a similar safety and immunogenicity profile to Shanchol vaccine, its effectiveness has not been ascertained in an outbreak situation in Zambia. The aim of this study, therefore, was to determine the effectiveness of Euvichol-plus vaccine during the 2017/2018 cholera outbreak in Lusaka district, Zambia. This evidence is important to support policy on the use of two doses of the Euvichol-plus vaccine as part of an integrated intervention to cholera control during outbreaks.

## METHODS

### Study design

This was a retrospective matched case–control study. A matched case–control design allows for controlling of confounding. It also allows the use of a relatively small sample, minimising the need for stratified analysis.[16]

### Study setting

The study was conducted in four (Chipata, Matero, Kanyama and Chawama) out of the six subdistricts of Lusaka which were affected by the cholera outbreak.

The combined population for the four subdistricts was 1 543 507.[17] The four districts were selected because that is where the 2018 Euvichol-plus vaccination campaign was conducted. For this purpose, the Zambian government procured a total of 2 070 100 doses of Euvichol-plus vaccine from (WHO) stockpile and implemented two rounds of the OVC campaigns. The first round was carried out from 10 to 20 January 2018, during which 1 317 925 people were vaccinated (as recorded in health facility vaccination reports). Due to logistical challenges, and while waiting for delivery of the complete vaccine stocks of OCV doses, the second round of the campaign was implemented in two phases. Phase 1 was conducted between 5 and 14 February 2018 and covered two of the four subdistricts (Chawama and Kanyama); phase 2 was implemented from 18 to 25 April 2018 and included the other two subdistricts (Chipata and Matero).

### Participants and procedures

During the 2017/2018 cholera outbreak, seven cholera treatment centres (CTCs) were set up in Kanyama, Chipata, Matero, Chawama, Bauleni, Kalingalinga subdistricts and a central CTC at the National Heroes Stadium. To aid and standardise management of cholera cases in the CTCs, a case definition was established and circulated to all the CTCs. Study participants included cases and controls from the CTCs in the four subdistricts. Cases were selected from the line list of suspected cholera patients admitted to the CTCs. A suspected cholera case was defined as any person who presented to the CTC with at least three episodes of watery diarrhoea within the last 24 hours, with or without dehydration or vomiting. On admission to the CTCs, patients suspected to be cholera cases submitted stool samples, which were sent to the microbiology laboratory at the University Teaching Hospital (in Lusaka) for confirmatory analysis using culture. A case was defined as any suspected cholera case with a culture positive result. Controls were recruited by trained research assistants with help from members of the research team. Controls were matched to cases by place of residence (neighbourhood), age and sex, and were systematically selected by visiting the fourth house to the right of a case's home and every consecutive house thereafter. In the event that a potential matched control participant was not available at the selected house at the time of the visit, an appointment was made to return to the house within 48 hours. If the participant was not found after two visits, the next non-enrolled house to the right was visited. Only one control was enrolled per household. If more than one person in a household satisfied the matched control criteria, the person closest in age to the case was enrolled as a control.

### Eligibility criteria

To be eligible for inclusion as a case in the study, a participant needed to:

▶ Give written informed consent.

- ► Have resided in the study area at the time of the vaccination campaigns.
- ► Have been at least 1 year of age or older at the time of the vaccination campaigns and had submitted faecal specimen that was found to be culture positive for *V. cholerae* O1.

Inclusion criteria for the control participants included

- ► Any person, aged 1 year or older at the time of the vaccination campaign.
- ► Residing in the study area at the time of the vaccination campaigns.
- ► No history of acute watery diarrhoea between January and June 2018.

## Sample size estimation

To determine the required sample size for the evaluation and analysis of the Euvichol-plus vaccine effectiveness, the following assumptions were made: (1) 80% of the target population received two doses of OCV; (2) two doses of the vaccine were 75% effective, (3) the study would have 80% power and (4) 5% margin of error. Based on these assumptions, we determined that the study would need to enrol a minimum of 150 respondents (30 cases and 120 controls; ratio, 1:4). A total of 395 respondents (ie, 79 cases and 316 controls) were therefore recruited in the study and involved in the analysis.

## Variables

The study variables included the outcome and independent variables as follows:

### Outcome variable

cholera status among cases and controls.

### Independent variables

1. Demographic information: age, sex, number of children, place of residence.
2. Socioeconomic information: level of education, occupation, level of income.
3. Vaccination status.

## Data collection procedures

An electronic questionnaire, loaded onto a handheld electronic tablet using the Open Data Kit (ODK) application was used as a data collection tool. Before administering the questionnaire, the research assistant explained the purpose of the study to the participants. Next, informed consent (patient consent form) was obtained from the participants who agreed to participate in the study. In addition, in the case of minors, ascent was obtained from guardians or parents. Next, verification of the vaccination exposure was done. To further ascertain vaccination status, participants were asked if they had been vaccinated during the two rounds of the 2018 campaigns and if so, to produce a confirmatory vaccination card. A participant's vaccination exposure was recorded as self-reported if a participant did not possess a vaccination card. Thereafter, data on demographic, socioeconomic and environmental variables (including type of toilet, source of water, level of education, income) was collected and responses recorded onto the ODK. Study participants were either coded zero, one or two dose vaccine exposure, based on their responses. Vaccine exposure was reclassified as zero dose if a participant reported having received a single dose but spat or vomited it within 1–2 hours of taking the vaccine, or developed cholera prior to receiving the first dose of vaccine. In addition, vaccine exposure status was reclassified as one dose if a study participant reported having received two doses of vaccine but developed cholera after the first dose was considered effective and prior to administration of a second dose. Multiple doses of vaccine administered to a study participant during the same vaccine campaign (7–10 days period) were treated as a single dose vaccine exposure in the analysis.

## Bias

Possible biases in the study include selection bias, measurement error and confounding due to non-response to vaccination and reinfection. To minimise selection bias, cases were those with a positive culture result, and selected from the line list in CTCs. Controls were matched with cases on residence (neighbourhood), age and sex. In addition, a group of research team members worked together to select the study sample. To minimise measurement error, research assistants were trained in data collection techniques. To minimise misclassification, a vaccination certificate/card was used to confirm vaccination status. Research assistants worked under supervision from the field supervisors. A pretested electronic data collection instrument, ODK was used. Confounding due to place of residence, age and sex were controlled through matching and conditional logistic regression analysis.

## Statistical analysis

Stata/SE V.14 (StataCorp)[18] was used for data analysis. To derive the household asset index, tetrachoric principle component analysis was performed. Descriptive statistics were done to summarise demographic and socioeconomic characteristics of participants (table 1). Next, bivariate analyses were carried out to compare and determine if there were differences in demographic and socioeconomic variables between cases and controls in order to identify potential confounders (table 2). Variables found to be significantly associated with case–control status (p<0.1) in the bivariate analysis required adjustment in the conditional logistic regression model (table 3). Conditional regression analysis was used to determine the association between vaccination status and case and control outcome, accounting for matching variables (place of residence, age and sex). Crude and adjusted OR (AOR) of being vaccinated among the cases and controls, 95% CI and vaccine effectiveness were computed. Vaccine effectiveness was calculated as (1−OR)×100. All reported p values and 95% CIs are two-sided (p<0.05).

**Table 1** Demographic characteristics

| Characteristic | Cases (n=79) | Controls (n=316) | P value | Total (n=395) |
|---|---|---|---|---|
| Age (years) | | | | |
| Mean (SD) | 20.19 (16.26) | 20.55 (16.16) | 0.85 | 20.47 (16.16) |
| Sex n (%) | | | | |
| Male | 45 (57%) | 186 (59%) | 0.76 | 231 (58.5%) |
| Female | 34 (43%) | 130 (41%) | | 164 (41.5%) |
| Education | | | | |
| None | 23 (29%) | 68 (22%) | 0.24 | 91 (23%) |
| Primary | 28 (35%) | 131 (41%) | | 159 (40%) |
| Secondary | 22 (28%) | 104 (33%) | | 126 (32%) |
| Tertiary | 6 (8%) | 13 (4%) | | 19 (5%) |
| Occupation | | | | |
| Formal employment | 6 (8%) | 18 (6%) | 0.32 | 24 (6%) |
| Informal Employment | 14 (18%) | 58 (18%) | | 72 (18%) |
| Unemployed | 12 (15%) | 54 (17%) | | 66 (17%) |
| House wife | 8 (10%) | 61 (19%) | | 69 (18%) |
| Student | 18 (23%) | 66 (21%) | | 84 (21%) |
| Child not in school | 21 (26%) | 59 (19%) | | 80 (20%) |
| Means of transport | | | | |
| Bicycle | 2 (3%) | 1 (0.3%) | 0.19 | 3 (1%) |
| Car | 23 (29%) | 81 (25.6%) | | 104 (26%) |
| Public transport | 13 (16%) | 66 (20.9%) | | 79 (20%) |
| Walking | 41 (52%) | 168 (53.2%) | | 209 (53%) |
| Average time taken to get to nearest health facility (min) | 20.39 (14.36) | 21.73(16.14) | 0.67 | 21.07 (15.53) |
| Household asset index | | | | |
| 0-4 | 36 (45%) | 105 (34%) | 0.28 | 141 (36%) |
| 5-7 | 20 (25%) | 87 (28%) | | 107 (28%) |
| 8-12 | 17 (22%) | 88 (29%) | | 105 (27%) |
| 12+ | 6 (8%) | 29 (9%) | | 35 (9%) |
| Number of household members | 6.19 (2.85) | 5.84 (2.39) | 0.39 | 5.95 (2.47) |
| Consumed food at the market | | | | |
| No | 44 (56%) | 202 (64%) | 0.18 | 246 (62%) |
| Yes | 35 (44%) | 114 36%) | | 149 38%) |
| Water source | | | | |
| Piped water | 57 (72%) | 256 (81.0%) | 0.27 | 313 (79%) |
| Well | 5 (6%) | 11 (3.5%) | | 16 (4%) |
| Borehole | 17 (22%) | 45 (14.3%) | | 62 (16%) |
| Rain water | 0 (0%) | 2 (0.6%) | | 2 (0.5%) |
| Bottled water | 0 (0%) | 2 (0.6%) | | 2 (0.5%) |
| Water treated before use | | | | |
| No | 36 (46%) | 139 (44%) | 0.8 | 175 (44%) |
| Yes | 43 (54%) | 177 (56%) | | 220 (56%) |
| Soap used when washing hands | | | | |
| No | 36 (46%) | 122 (39%) | 0.26 | 158 (40%) |
| Yes | 43 (54%) | 194 (61%) | | 237 (60%) |
| Type of toilet | | | | |

**Table 1** Continued

| Characteristic | Cases (n=79) | Controls (n=316) | P value | Total (n=395) |
|---|---|---|---|---|
| Private pit latrine | 13 (17%) | 63 (21%) | 0.2 | 76 (20%) |
| Shared pit latrine | 60 (79%) | 214 (70%) | | 274 (72%) |
| Toilet inside house | 1 (1%) | 22 (7%) | | 23 (6%) |
| Other | 2 (3%) | 6 (2%) | | 8 (2%) |

## Patient and public involvement

The study design was determined by the research team. Participants and the public were not directly involved in the conceptualisation and design of the study. However, selection of the study sites was done in consultation with stakeholders from the MoH and Zambia National Public Health Institute. Selection of study participants was done in collaboration with the provincial and district health managers. A dissemination meeting was held in Lusaka and study findings shared with key stakeholders, including the WHO, MoH and community leaders in the health facilities where the study was conducted. A final report was also written and shared with the funding organisation.

## RESULTS
### Participants

A summary of the recruitment algorithm of study participants is shown in figure 1. A total of 5715 patients were recorded from the six CTCs in Lusaka district, from which 2761 (48.3%) stool specimens were collected and sent for laboratory confirmation. Of the total number of samples sent for laboratory confirmation, 931 (33.7%) were culture positive for *V. cholerae* O1. Out of the 931 culture positive cholera cases, only 265 (28.5%) cases had verifiable contact information and were contacted for possible participation in the study; 97 (10.4%) cases voluntarily agreed and accepted to be interviewed and enrolled in the study. These were matched to 388 controls. On review of the data, cases and controls were found ineligible and did not meet the inclusion criteria (age, sex and vaccine status) were excluded from the analysis. Matched cases and controls were excluded from the analysis if their vaccine exposure could not be verified. A study participant's vaccine exposure could not be verified if they did not know whether they had received a dose of the vaccine or not. Cases were also excluded if they developed cholera during the 10 days after receiving the vaccine in the campaign. Thus, 18 cases and their corresponding 72 matched controls did not meet the inclusion criteria (age, sex and unknown vaccine status) and were excluded, leaving a total of 79 cases and 316 controls (n=395) in the final analysis (figure 1).

## Demographic characteristics of the study participants

The demographic characteristics of the study participants are shown in table 1. Overall, majority of the respondents were male (58.5) with a mean age of 20.19 (16.26) years for the cases and 20.55 (16.16) years for the controls. Majority of participants (72%) had either primary or secondary education. Most study participants (64%) were classified in the lower two quartiles of the generated household asset index. The average household size was 6 (mean=5.95, SD=2.47) persons. Almost four in five (79%) study participants used piped water as their primary water source; only 56% treated their drinking water. Most (72%) used a shared pit latrine and 60% reported using soap during handwashing. On average, study participants lived 21.07 (15.53) minutes walk from the nearest health facility and the most commonly (53%) reported mode of transportation was walking (53%). Less than half (38%) of the study participants reported having consumed food from a market during the outbreak.

## Vaccination status

Out of the total number of participants included in the analysis, 20 (25.3%) cases and 42 (13.3%) controls were unvaccinated (zero dose). Of the total sample, 49 (62.0%) cases and 79 (25%) controls received one dose; 10 (12.7%) cases and 189 (59.8%) controls received two doses, and 59 (74.7%) cases and 268 (84.8%) controls were classified as having received any dose of vaccine (either one or two doses).

## Bivariate analysis

Bivariate analysis showed no significant differences between the cases and controls with regard to sex, occupation, means of transport and drinking treated water. Significant differences between the two groups were noted with regard to education, number of people in the households (table 2).

## Conditional logistical regression analysis

Conditional logistical regression analysis showed a significant association between two doses of the Euvichol-plus OCV and vaccine protection (AOR=0.19; 95% CI 0.16 to 0.28) with a vaccine effectiveness of 81% (95% CI 72.0% to 84.0%; p value <0.01) (table 2). The effectiveness of any (one or more) doses of Euvichol-plus vaccine was 74% (95% CI 50.0% to 86.0%; p value <0.01) (table 3).

## DISCUSSION

The aim of this study was to determine the effectiveness of Euvichol-plus vaccine administered during a mass OCV campaign as part of control measures following a cholera outbreak in Lusaka. Our findings show that two

**Table 2** Vaccination status

| | Vaccination | | | | | |
| --- | --- | --- | --- | --- | --- | --- |
| | Case (n=79) | | | Control (n=316) | | |
| | Yes=59 | No=20 | P value | Yes (n=274) | No=42 | P value |
| Variable | n (%) | n (%) | | n (%) | n (%) | |
| Sex | | | 0.594 | | | 0.087 |
| Male | 34 (57.6) | 13.0 (65.00) | | 166 (60.7) | 31 (73.8) | |
| Female | 25 (42,4) | 7.0 (35.0) | | 108 (39.3) | 11 (26.2) | |
| Age group | | | 0.032 | | | 0.001 |
| <5 | 19 (32.1) | 1 (5.0) | | 77 (28.1) | 5 (11.9) | |
| 5–9 | 9 (15.3) | 1 (5.0) | | 40 (14.6) | 2 (4.8) | |
| 10–19 | 6 (10.2) | 1 (5.0) | | 23 (8.4) | 2 (4.8) | |
| 20–29 | 10 (16.9) | 5 (5.0) | | 46 (16.8) | 16 (38.0) | |
| 30–39 | 9 (15.3) | 7 (5.0) | | 49 (17.9) | 11 (26.2) | |
| 40+ | 6 (10.2) | 5 (25.0) | | 39 (14.2) | 6 (14.3) | |
| Occupation | | | 0.021 | | | 0.065 |
| Child not in school | 17 (28.8) | 1.0 (5.0) | | 50 (18.3) | 7 (16.7) | |
| Formal employment | 5 (8.5) | 4 (20.0) | | 17 (6.2) | 6 (14.3) | |
| Informal employment | 6 (10.2) | 7 (35.0) | | 50 (18.3) | 13 (31.0) | |
| Housewife | 7 (11.8) | 1 (5.0) | | 51 (18.6) | 3 (7.1) | |
| Student | 14 (23.7) | 2 (10.0) | | 53 (19.3) | 5 (11.9) | |
| Unemployed | 6 (10.2) | 4 (20.0) | | 51 (18.6) | 8 (19.0) | |
| Other | 4 (6.8) | 1 (5.0) | | 2 (0.7) | 0 | |
| Education | | | 0.037 | | | 0.000 |
| None | 26 (33.8) | 2 (10.0) | | 56 (20.5) | 8 (19.1) | |
| Primary | 30 (39) | 6 (30.0) | | 119 (43.4) | 7 (16.7) | |
| Secondary | 16 (20.8) | 9 (45.0) | | 89 (32.4) | 21 (50) | |
| Tertiary | 5 (6.4) | 3 (15.0) | | 10 (3.7) | 6 (14.2) | |
| Number of people in household | | | 0.694 | | | 0.032 |
| 0–4 | 16 (27.1) | 7 (35.0) | | 79 (28.8) | 20 (47.6) | |
| 5–7 | 26 (44.1) | 9 (45.0) | | 136 (49.6) | 14 (33.3) | |
| 8–12 | 14 (23.7) | 4 (20.0) | | 55 (20.1) | 8 (19.1) | |
| 12+ | 3 (5.1) | 0.0 | | 4 (1.5) | 0 (0.0) | |
| Means of transport to health facility | | | 0.199 | | | 0.071 |
| Bicycle | 1 (1.7) | 1 (5.0) | | 0 | 1 (2.4) | |
| Walking | 35 (59.3) | 7 (35.0) | | 143 (52.2) | 23 (54.7) | |
| Car | 14 (23.7) | 8 (40.0) | | 70 (25.5) | 11 (26.2) | |
| Public transport | 9 (15.3) | 4 (20.0) | | 61 (22.3) | 7 (16.7) | |
| Drinking treated water | | | 0.281 | | | 0.506 |
| Yes | 34 (57.6) | 9 (45.0) | | 156 (56.9) | 22 (52.4) | |
| No | 25 (42.4) | 11 (55.0) | | 118 (43.1) | 20 (47.6) | |

doses of Euvichol-plus vaccine confer effective protection against cholera and can thus serve as an intervention in the control of cholera outbreaks.

These findings add to the existing evidence on the effectiveness of a two-dose regimen of OCV. They are also consistent with previous studies, for example, a randomised control trial conducted in the Philippines[16] comparing Euvichol and WC-rBS showed that the efficacy of two doses of Euvichol-plus vaccine was non-inferior to that of WC-rBS in adults (80% vs 74%) and children (91% vs 88%).[16] Thus, our findings suggest that two doses of Euvichol-plus OCV administered during

**Table 3** Effectiveness of Euvichol vaccination

| Number of doses | Cases (n=79) | Controls (n=316) | AOR* (95% CI) | VE | P value |
|---|---|---|---|---|---|
| Any dose | 51 | 268 | 0.26 (0.14 to 0.50) | 74.0% (50–86) | <0.01 |
| 2 | 10 | 189 | 0.19 (0.16 to 0.28) | 81.0 (72–84.0) | <0.01 |

*Adjusted for age, education, place of residence.
VE, vaccine effectiveness.

a mass vaccination campaign in response to a cholera outbreak confer effective protection. This finding is also in line with previous observational studies that showed that killed OCVs are protective in endemic and outbreak settings.[19 20] In an effectiveness study of mass OVC conducted in Beira, Mozambique, Lucas and colleagues[21] found that two doses of WC-rBS vaccine conferred 78% (95% CI 29% to 92%) protection when administered during the outbreak. These findings are also consistent with those reported from Tanzania where the WC-rBS vaccine conferred a protection of 79% (95% CI 47% to 92%) when administered in cholera endemic areas of Zanzibar.[22] Similarly, two doses of BivWC OCV was found to be effective when deployed in response to cholera outbreaks and areas of high endemicity in Haiti, India and Guinea, where they conferred 63% (95% CI 8% to 85%), 69% (95% CI 15% to 89%) and 87% (95% CI 57 to 96%), respectively.[23]

Our findings also show that administration of any dose (one or two) conferred significant protection against cholera. This finding is consistent with previous observational studies which reported the effectiveness of a single dose of WC-rBS vaccine.[24 25] In South Sudan, for example, a study showed that the short-term effectiveness of a single dose of WC-rBS vaccine was 87%, when administered during outbreaks. Similarly, a study conducted in Haiti reported a 79% effectiveness (95% CI 43% to 93%) of one dose of OCV.[26]

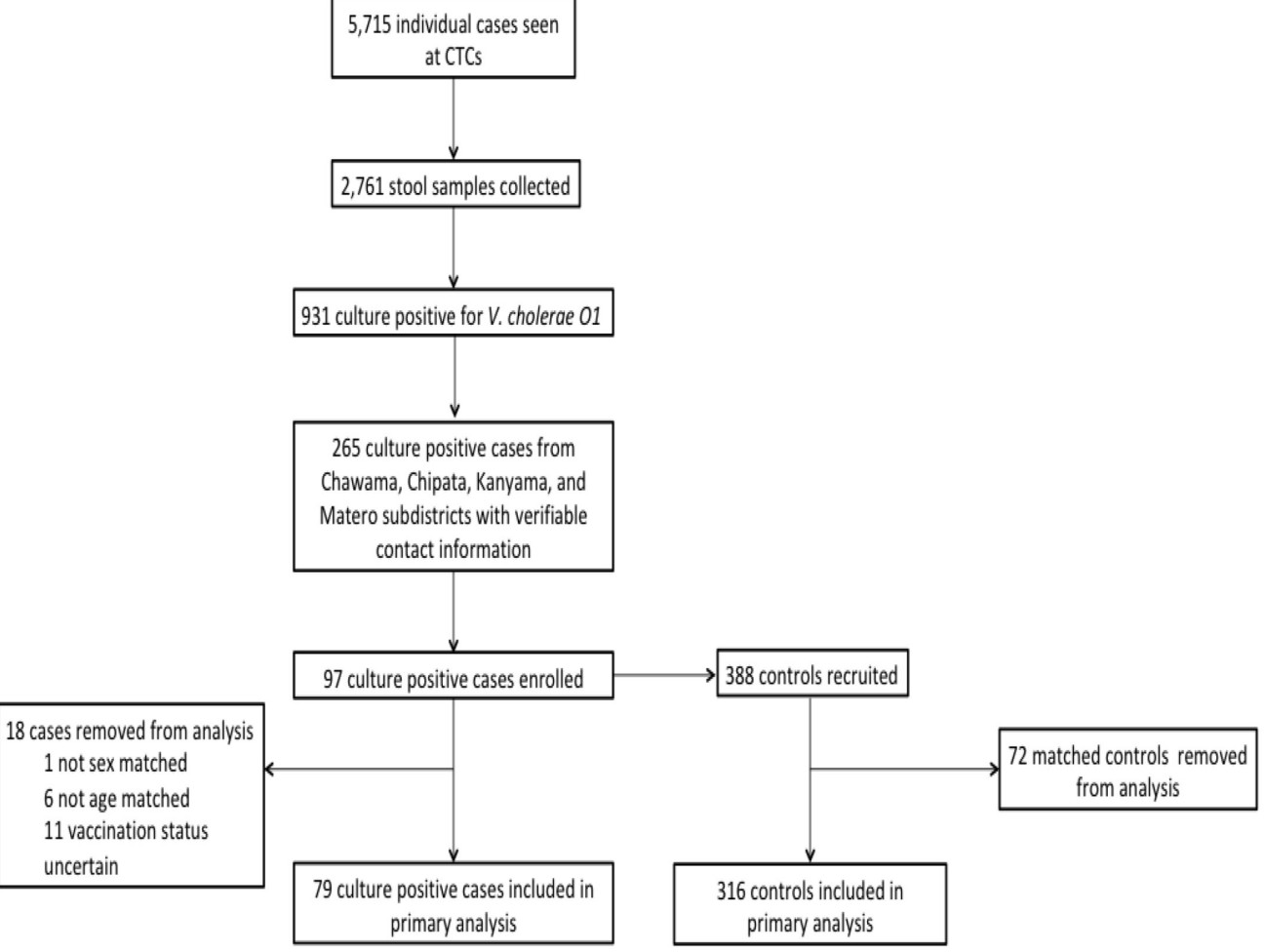

**Figure 1** Participant recruitment algorithm. CTC, cholera treatment centre.

Despite these benefits, our study did not have enough power to determine the long-term effectiveness of the two doses, and thus, it is unclear how long these doses would confer protection against cholera. Nevertheless, available evidence shows that vaccine effectiveness wanes over time.[26] A systematic review on the protection against cholera from killed whole-cell OVCs conducted by Bi and colleagues and published in the *Lancet* found that, although the average effectiveness of two doses of OCV at 1-year postvaccination was 83%, it decreased to 69% after 2 years. Another study comparing the long-term vaccine effectiveness of two doses against one dose of OCV conducted in Haiti reported that two doses have a significantly higher long-term effectiveness than one dose.[25] It further showed a 76% protection up to 4 years for two dose vaccine.[26] On the contrary, these authors found that the 79% effectiveness of one dose of OCV observed at 1 year was completely diminished after 2 years. Thus, these findings clearly show the benefits of administering two doses of OCV. However, further research employing longitudinal study design is required to determine the long-term effectiveness of two doses of OCV in the local context.

Potential limitations of this study should be noted. First, our study findings on the effectiveness of two doses are based on a small sample (10 cases and 189 controls) that received two doses of the Euvichol-plus vaccine. Thus, the study could have been underpowered. Second, our OCV effectiveness was based on an observational epidemiological study design following a reactive vaccination campaign mounted in response to a cholera outbreak; evidence from observational studies is considered relatively weak. Third, the vaccine exposure status was based on participant self-reports and cards for both cases and controls. This could have introduced information bias and misclassification. In addition, selection of cases from the line list of all suspected cholera patients treated in various CTCs between January and June 2018 might not have represented cholera cases that did not seek care at a health facility.

Despite these limitations, our findings are important as they add to the current evidence on the effectiveness of a two-dose regimen of OCV in the control of cholera outbreaks. In addition, our use of a matched case–control study design, collection of data shortly after the OCV campaign and conditional regression analysis all minimised confounding and increased validity of the study findings. Moreover, selection of cases from the line list with confirmed diagnosis reduced misclassification bias, making the study findings valid.

## CONCLUSION

Our findings show that two doses of Euvichol-plus vaccine are effective in the control of the cholera outbreak in Lusaka, Zambia. The findings also show that two doses of the vaccine are more effective than a single dose. These results provide the evidence for and support the use of two doses of the vaccine as part of an integrated intervention to cholera control during outbreaks. Nevertheless, it is unclear how long these doses would confer protection against cholera and the long-term effectiveness of the two doses is not well understood. Longitudinal studies are required to determine the long-term effectiveness of two doses of the OCV among the vaccinated populations in the local context. Further research is also required to determine the effectiveness and usefulness of Euvichol-plus vaccine in conferring herd immunity among non-vaccinated individuals during mass immunisation and to determine the required minimum coverage. Finally, further research is needed to determine Euvichol-plus vaccine effectiveness among people living with HIV and its usefulness among these populations.

**Author affiliations**
[1]School of Public Health, Levy Mwanawasa Medical University, Lusaka, Zambia
[2]Zambia National Public Health Institute, Lusaka, Zambia
[3]Surveillance and Disease Intelligence, Zambia National Public Health Institute, Lusaka, Zambia
[4]Ministry of health Zambia, Lusaka, Zambia
[5]World Health Organization, Lusaka, Zambia
[6]Disease Surveillance, World Health Organisation, Lusaka, Zambia
[7]Communication Information & Research, Zambia National Public Health Institute, Lusaka, Zambia
[8]University of California Berkeley, Berkeley, California, USA
[9]Laboratory System and Networks, Zambia National Public Health Institute, Lusaka, Zambia
[10]Laboratory Systems and Networks, Zambia National Public Health Institute, Lusaka, Zambia
[11]Tropical Diseases Research Centre, Ndola, Zambia
[12]University of Zambia School of Public Health, Lusaka, Zambia
[13]Epidemic Preparedness & Response, Zambia National Public Health Institute, Lusaka, Zambia
[14]Zambia Ministry of Health, Lusaka, Zambia
[15]The Copperbelt University School of Medicine, Kitwe, Zambia

**Acknowledgements** We thank Lusaka District Health Management Team who provided field research assistants and study participants. The GTFCC supported the country with OCV stocks and operations funds.

**Contributors** VMM, CS, OC, BBM, AMN, BG, WN, MKapina, LM and MKasonde contributed to the conception of the study, literature search, data collection as well as the drafting of the manuscript including data analysis, figures and results interpretation. KM, CM, NS, PB, KZ, LM and MM contributed to the coordination of the field study activities. VMM, BBM, NB, KM, CM contributed to the coordination of conception of the study and the literature search. CS, MK, BBM, WN, MM, NB contributed to the revision of the manuscript. All authors read and approved the manuscript. VMM acted as the guarantor for the study.

**Funding** Ministry of Health and WHO.

**Competing interests** None declared.

**Patient and public involvement** Patients and/or the public were involved in the design, or conduct, or reporting, or dissemination plans of this research. Refer to the Methods section for further details.

**Patient consent for publication** Consent obtained directly from patient(s)

**Ethics approval** This study involves human participants and was approved by Excellence in research ethics and science (ERES) Converge IRB (Ref: ERES/2019-09-001). Participants gave informed consent to participate in the study before taking part.

**Provenance and peer review** Not commissioned; externally peer reviewed.

**Data availability statement** Data are available upon reasonable request. Data are available upon reasonable request from the corresponding author and with permission of the ERES ethics review board.

**ORCID iDs**
Cephas Sialubanje http://orcid.org/0000-0002-9077-1436
Albertina Moraes Ngomah http://orcid.org/0000-0002-3435-1170
Modest Mulenga http://orcid.org/0000-0002-0551-3919

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
