## [Reviewer comments · BMJ Open]

ARTICLE DETAILS

TITLE (PROVISIONAL)	Effectiveness of two doses of Euvichol-plus oral cholera vaccine in response to the 2017/2018 outbreak: A matched case control study in Lusaka, Zambia
AUTHORS	Sialubanje, Cephas; Kapina, Muzala; Chewe, Orbrie; Matapo, Belem; Ngomah, Albertina; Gianetti, Brittany; Ngosa, William; Kasonde, Mpanga; Musonda, Kunda; Mulenga, Modest; Michelo, C; Sinyange, Nyambe; Bobo, Patricia; Zyambo, Khozya; Mazyanga, Lucy; Bakayita, Nathan; Mukonka, Victor M

VERSION 1 – REVIEW

REVIEWER	von Seidlein, Lorenz Menzies School of Health Research
REVIEW RETURNED	04-Sep-2022

GENERAL COMMENTS	This is a well-designed and reported study. The writing is exemplary. I have only minor suggestions: Please add to the weaknesses of the study design reported on page 3: 1. an observational study is not a randomised controlled trial. experts tend to consider evidence from such case control studies as second rate.2. by necessity the study is not blinded3. no bias indicator study was conducted Figure selection 2,761 samples from 5,715 cases not clear why and how this subset was selected. (apologies if I overlooked something ...) Figure selection 388 controls recruited why and how was the subset of 316 controls selected for analysis. (apologies if I overlooked something ...) P7 L45 eligibility criteria. Residing in the study area – should also include the focal time i.e. at the time of the vaccination campaigns? Please explain why twice as many cases were enrolled as needed in the sample size estimation? 'Based on these assumptions, we determined that the study would need to enroll a minimum of 150 respondents (30 cases and 120 controls; ratio, 1:4). A total of 395 respondents (that is, 79 cases and 316 controls) were therefore recruited in the study and involved in the analysis.' Abstract – last word 'out breaks' should read outbreaks. P7 L14 'Controls were recruitment' should read Controls were recruited
---

REVIEWER	Ray, Arindam Bill and Melinda Gates Foundation India
REVIEW RETURNED	05-Oct-2022

GENERAL COMMENTS	Well conducted study with due elaboration of limitations.
---

VERSION 1 – AUTHOR RESPONSE

Reviewer: 1

Query 1: An observational study is not a randomised controlled trial; experts tend to consider evidence from such case control studies as second rate and that 1) by necessity the study is not blinded; 2) no bias indicator study was conducted

Response: We appreciate the comment from the reviewer; we have added this weakness accordingly (see page 3)

Query 2a: The reviewer observed that in the selection figure, 2,761 samples were selected from 5,715 cases and that it was not clear why and how this subset was selected. The reviewer also observed that in the same figure, 388 controls were recruited and wondered why and how the subset of 316 controls was selected for analysis.

Response: We value the observation by the reviewer. We explained that out of a total of 5,715 patients recorded from the six cholera treatment centres (CTCs) (in Lusaka district, only 2,761 (48.3%) stool specimens were collected and sent for laboratory confirmation. This analysis is based on the records kept at the CTCs. Since this is an observational and not interventional study, it is not clear why stool specimens were not collected from the other patients. We have taken note of this gap in the limitation section (see page 9)

Query 2b: The reviewer also observed that in the selection figure, 388 controls were recruited and wondered why and how the subset of 316 controls was selected for analysis.

Response: We appreciate this observation. We have clarified the selection process accordingly (see page 9)

Query 3: The reviewer advised that on P7 L45 eligibility criteria, residing in the study area should also include the focal time i.e. at the time of the vaccination campaigns

Response: We have edited this section accordingly (see page 6)

Query 4: Please explain why twice as many cases were enrolled as needed in the sample size estimation? 'Based on these assumptions, we determined that the study would need to enroll a minimum of 150 respondents (30 cases and 120 controls; ratio, 1:4). A total of 395 respondents (that is, 79 cases and 316 controls) were therefore recruited in the study and involved in the analysis.'

Response: We appreciate the concern by the reviewer. As pointed out, we computed the minimum sample size required for the study. However, as a general rule, increasing the sample size (above the minimum) would increase the power of the study.

Query 4: Abstract – last word 'out breaks' should read outbreaks; P7 L14 'Controls were recruitment' should read Controls were recruited.

Response: The corrections have been made accordingly